# Strength and Expansion of LHEC with Different Gypsum Contents Under Thermal Curing

**DOI:** 10.3390/ma17235766

**Published:** 2024-11-25

**Authors:** Bingxin Jin, Shuanglei Wu, Shujing Fan, Fafu Hang, Huxing Chen

**Affiliations:** 1School of Materials Science and Engineering, Zhejiang University, Hangzhou 310027, China; 22226005@zju.edu.cn (B.J.); wushuanglei@zju.edu.cn (S.W.); 2Linhai Zhongxin New Building Materials Co., Ltd., Taizhou 317000, China; liuluorenjiantree@163.com (S.F.); hanghuohuo181030@163.com (F.H.)

**Keywords:** low-carbon cement, LHEC, gypsum content, temperature, strength, expansion

## Abstract

Low-heat expansive cement (LHEC) is an environmentally friendly and low-carbon cementitious material. Compared to ordinary Portland cement (OPC), LHEC reduces CO_2_ emissions from the cement production process; furthermore, it enhances the service life of the cement by overcoming the problem of OPC’s strength inversion in hot and humid environments. In order to improve the performance of LHEC in a hygrothermal environment, the strength and expansion of LHEC with different gypsum dosages (8–20%) at curing temperatures of 20 °C, 50 °C, and 80 °C were investigated. The corresponding mechanism was investigated using XRD, TGA, SEM, and porosity analyses. The results indicate that there is a ‘critical gypsum dosage’ for strength at 20 °C. The ‘critical dosage’ rises with the curing temperature or an increase in age. Raising the curing temperature has a better effect on the strength of cement with a higher gypsum dosage; it does not have such a positive effect on cement with a low gypsum dosage. The higher the gypsum content, the greater the expansion rate, and the longer the time needed for the expansion to stabilize. The higher the curing temperature, the shorter the time required for stable expansion and the lower the final expansion rate. Increasing the gypsum dosage and maintaining the temperature promote the hydration of slag and the formation of ettringite (AFt), thereby enhancing the microstructure of the cement. AFt decomposition occurs in the case of a low gypsum dosage and high curing temperature. According to the above results, it is inferred that the strength and expansion performance of LHEC in a hygrothermal environment can be improved by appropriately increasing its gypsum dosage. This finding offers valuable insights for the improvement of LHEC and its application in hygrothermal conditions.

## 1. Introduction

Large quantities of cement are consumed globally each year. Due to its contributions to pollution, resource depletion, and greenhouse gas emissions, cement has a significantly negative influence on the environment. China is a major cement producer, producing 2.4 billion tons of cement annually. The production of ordinary Portland cement (OPC), which is prepared using limestone that is calcined at high temperatures, is the main cause of CO_2_ emissions; the construction industry contributes about 4 to 6% of global CO_2_ emissions [1]. A single ton of ordinary Portland cement (OPC) demands around 4.0 G Joules of energy and creates about 0.7 tons of CO_2_ emissions. The Chinese cement industry emits 1.23 billion tons of CO_2_, accounting for about 13% of the country’s total carbon emissions. China is striving to achieve its goal of reducing carbon emissions, and reducing carbon emissions from the cement industry is an important historical task. Therefore, the promotion of green and low-carbon cement is an important issue. Many studies have explored methods to increase the sustainability of cement, such as the use of supplementary cementitious materials (SCMs) that have a positive influence on the mechanical properties and durability of cement [2] and the integration of plastic waste into building materials to reduce the consumption of natural aggregates [3].

Cementitious materials are applied in a wide range of applications; they are commonly used in scenarios in which they are exposed to hot and humid environments for long periods of time. Exothermic heat generated by radioactive decay can lead to high temperatures in cement repositories over time. Due to the exothermic heat generated by radionuclide decay, cement repositories are at a high temperature for a long time. The temperature of the rock surrounding repositories can reach about 80 °C. In mining engineering or transportation engineering, when geothermal or hot springs are encountered, the cement materials used are also exposed to a high temperature for a long time. The temperatures of the borehole wall of the diversion power tunnel of the Bulunkou-Gonger Hydropower Station in the Kunlun Mountain region of China and the rock wall of the diversion tunnel of the Qizhehatar hydropower station in Xinjiang, which is nearly one kilometer in length, have been observed to reach more than 80 °C. In mass concrete, due to the large cross-section, a temperature gradient also occurs when the internal temperature rises sharply due to the hydration heat of the cement after the concrete is poured [4]. When simulating the hydration temperature field of cement abutments using the finite element method (FEM), Yonghui Huang discovered that the temperature of cement rises rapidly but decreases slowly and that the maximum temperature at the center of the cement reaches 86.6 °C [5]. In this case, the concrete exhibits tensile stress, and when the ultimate tensile strength is less than the tensile stress to which it is subjected, the mass concrete is highly susceptible to temperature cracks [6].

When applied in hygrothermal environments, the properties of OPC are not as good as they are in ambient environments; its hydration products are mainly C–S–H, CH, AFt, and monosulphoaluminate (AFm). The hydration products of OPC are similar in hot and humid ambient environments. The increased temperature promotes hydration, leading to high early strength [7,8]. In the case of increased temperature, however, this fast hydration in the initial stage leads to a more heterogeneous distribution of the hydration products as the hydrates precipitate around the clinker particles and build up a dense inner shell around the clinkers [9]. Meanwhile, the densities of C-S-H gel increase at higher temperatures, leading to a stronger coating effect on the unhydrated clinker, accompanied by a decrease in volume, which increases the porosity of the cement stone [10]; this results in quality problems, such as the strength shrinkage of ordinary Portland cement-based materials in a high-temperature environment [11].

Low-heat expansive cement (LHEC) is a cementitious material made of granulated blast furnace slag, which is the main component, to which appropriate amounts of Portland cement clinker and gypsum are added. Slag is an industrial waste product from pyrometallurgical processing; it can partly replace OPC and has a positive effect on the strength and durability of concrete [12]. Thus, the carbon emissions of LHEC are much lower than those of OPC. Promoting the application of LHEC to replace OPC is an important means of reducing carbon emissions in the cement industry. The slag content of LHEC is between that of supersulfated cement and slag Portland cement, which ensures a good strength performance; it also has low hydration heat and micro-expansion performance. There is a national standard (low-heat expansive cement, GB/T 2938-2008) [13] in China that has been issued for LHEC.

LHEC is widely used in large-scale projects, such as dams, due to its low hydration heat and micro-expansion properties. Unlike OPC, the main hydration product of LHEC is AFt, along with C-(A)S-H, and a small amount of CH. This cement has slightly lower early strength at room temperature, but higher strength at later stages. It has also demonstrated outstanding strength and expansion properties when applied in mass concrete. Based on the composition and performance characteristics of LHEC, it is expected that it will be more suitable for application in a hygrothermal environment than OPC. However, there are few studies on the hydration and strength properties of this cement under hygrothermal conditions. In this study, a consistent dataset on the strength and expansion of LHEC with different gypsum dosages (8–20%) at 20 °C, 50 °C, and 80 °C is presented.

The formation and stability of ettringite during hydration are critical to the performance of LHEC. At the molecular level, the curing temperature affects the crystal cell parameters, crystal nucleus size, and morphology of ettringite. At the macro level, it affects the solubility and stability of ettringite. It is generally accepted that the stable temperature range of ettringite is 40~50 °C, and the amount of ettringite increases with an increase in temperature in this range [14]. When the temperature reaches 70 °C, the formation of ettringite is accompanied by the partial decomposition of ettringite [15]. Beyond 80 °C, ettringite is mainly decomposed, and the amount of ettringite gradually decreases with an increase in temperature [16,17]. However, this is not absolute. The SO_4_^2−^ content in these studies is mostly low; thus, the AFt mostly exhibits dissolution, adsorption, or conversion to biotite, AFm [18]. It has been found that the incorporation of gypsum into Portland cement can promote the formation of ettringite and reduce the conversion of ettringite into AFm, thus lowering the reduction in strength and the increase in porosity caused by the temperature rise [19]. In the LHEC system, gypsum is a key component, which not only determines the amount and stability of AFt formation in the hydration process but can also be seen as an exciter of slag activity, promoting the disintegration and hydration of slag. Therefore, appropriate gypsum content is important for the performance of LHEC when applied in hot and humid environments. Thus, increasing the gypsum content may also have a positive influence on the strength of LHEC curing at high temperature, because it increases the stability of AFt.

This study focused on investigating the influence of gypsum content on the strength and expansion properties of LHEC at 20 °C, 50 °C, and 80 °C, aiming to provide theoretical guidance for the improvement and application of this cement in the hygrothermal environment. This study is significant because it broadens the application situations for LHEC and improves its durability in the hygrothermal environment, in order to reduce the carbon emissions of the cement industry and promote the sustainable development of society.

## 2. Experimental Program

### 2.1. Materials and Proportion

Cement mortar samples were obtained from Portland cement clinker, granulated blast furnace slag (GBFS), and desulfurization gypsum. The Portland cement clinker used in this experiment was produced by a cement factory in Zhejiang Province, with a specific surface area of 360 m^2^/g. The GBFS used in this experiment was produced by Shanghai Baoshan Iron and Steel Company with a grade of S 95 and a specific surface area of 415 m^2^/g. The gypsum in this experiment was desulfurization gypsum produced by Zhejiang Xindu Cement Company. The chemical compositions of the materials are shown in Table 1. The compositions of the mortars are summarized in Table 2.

### 2.2. Strength Test

The mortar specimens were mixed in a mechanical mixer and cast in cuboid molds of 40 mm × 40 mm × 160 mm, in accordance with GB/T 17671-2021 [20]. The sand-to-binder ratio was equal to 3:1. The w/b was kept constant at a value of 0.5. Mortar specimens were demolded after molding into standard curing box for 24 h. Then, the hardened specimens were cured in water at a constant temperature and humidity in a standard cement curing box at 20 °C and at a constant water temperature in an electrothermal temperature box, at two temperatures (50 and 80 °C), respectively, for 7 d, 28 d, 60 d, and 120 d. Flexural and compressive strength tests were performed when the specimens were cured to a specified age. The average flexural strength of three specimens is reported. The compressive strength was determined using the broken halves of the specimens. The reported value of the compressive strength is the mean of six tests. The images of the experimental tests are shown in Appendix A.

### 2.3. Expansion Test

The expansive specimens were cast in cuboid molds of 25 mm × 25 mm × 280 mm, with two copper nail heads at both ends of the long axis of the specimens. The w/b was kept constant at a value of 0.28. After molding, the specimens were demolded into a standard curing box for 48 h, and the initial length $L_{0}$ was tested immediately after demolding. Then, the hardened specimens were cured at a constant temperature and humidity in a standard cement curing box at 20 °C and at a constant water temperature in an electrothermal temperature box, at two temperatures (50 and 80 °C), respectively. During curing, the lengths $L_{x}$ of the specimens were tested at certain ages. The expansive ratio of the specimens ($E_{x}$) is demonstrated in Equation (1):(1)$$E_{x}=\frac{L_{x}-L_{0}}{280}\times 100$$

### 2.4. XRD Analysis

The crystalline phases of the hydration products were characterized by employing an X-ray diffractometer (XRD, Bruker D8 advance, Karlsruhe, Germany). The powdered samples were scanned between 5° and 80° at a rate of 4°/min.

### 2.5. TG Analysis

TG analysis was carried out on 20 ± 2 mg of the powdered samples using a Mettler Toledo TGA/DSC3+ Instrument under a nitrogen atmosphere. The samples were heated from ambient temperature to 1000 °C at a ramp rate of 10 °C/min. The DTG curve was obtained after derivation to demonstrate various peaks corresponding to the dehydration of different phases.

### 2.6. SEM Analysis

A relatively flat and thinly sliced sample was cut out with a knife and treated with gold plating; then, it was used for the observation of surface morphology using scanning electron microscopy (SEM, FEI FEG650, Shanghai, China). An accelerating voltage of 20 kV and a working distance of 10.3 mm were used.

### 2.7. MIP Analysis

Block samples with sizes of less than 1 × 1 × 1 cm were selected for pore structure analyses. The pore structure of the fragmented samples was evaluated by means of a mercury intrusion porosimeter (MIP, Micro Active Auto Pore V 9600, Norcross, GA, USA). The measured pore sizes ranged from 5 nm to 0.8 mm.

## 3. Results

### 3.1. Strength

#### 3.1.1. Flexural Strength

The flexural strengths of the mortar specimens with varied gypsum contents cured at different temperatures in water were recorded at 7 d, 28 d, 60 d, and 120 d (Figure 1), using the mortar flexural strength test. As shown in Figure 1, the flexural strength of the specimen at 20 °C increased and then decreased with an increase in gypsum content. At 7 days, the flexural strength of the specimen at 20 °C reached the highest level when the gypsum content was 11%, which was 6.6 MPa. When the specimens were cured for 28 and 60 days, they also reached the maximum at 11%; the strength of the specimens with 14% dosing was slightly lower than that. When the gypsum content exceeded 14%, the strength of the specimen was significantly reduced. By 120 days, the flexural strength tended to grow with an increase in the curing temperature. It can be seen that there was a ‘critical dosage’: the strength increased or changed slightly with the gypsum when it was less than the dosage and decreased significantly when it was above it. The ‘critical dosage’ increased with an increase in the curing temperature or age.

When the specimens were cured at 50 °C, the rates of strength development at 7 and 28 days were similar to those at 20 °C, while the ‘critical dosages’ of the specimens were both 14%. The flexural strengths of the specimens at 60 d were basically no longer reduced and were both at 12 ± 1 MPa. At the age of 120 days, the strength of the specimens increased with an increase in gypsum content. Consequently, it can be concluded that the ‘critical dosages’ of the specimens were greater than 20% at 60 and 120 days.

When the curing temperature rose to 80 °C, the strength of the specimens at all ages increased with an increase in gypsum dosage; consequently, it can be concluded that the ‘critical dosages’ of the specimens were greater than the maximum dosage of 20%. Noticeably, although the strengths of the specimens at 50 °C and 80 °C were higher than those of the specimens at 20 °C, it was not the case that the flexural strength of the specimens improved with an increase in temperature. The effect of the curing temperature on the flexural strength was related to the gypsum content. When the gypsum content was lower than 14%, the flexural strength of the specimens at 50 °C was the highest, and when the gypsum content was higher than 14%, the flexural strength of the specimens at 80 °C was the highest.

#### 3.1.2. Compressive Strength

The compressive strengths of the mortar specimens (Figure 2) were determined using the mortar compressive strength test. As shown in Figure 2, the higher the curing temperature, the higher the compressive strength of the specimen when the same gypsum doping amount was used, except for S1 (with 8% gypsum content) when cured for 7 d. When cured at 20 °C, the strength of the specimen increased first and then decreased with an increase in gypsum content. As with the flexural strength, there was a ‘critical dosage’. When the specimens were cured for 7 days, the compressive strengths of the specimens reached the highest (28.5 MPa) when the gypsum doping amount was 11%. The compressive strength of the specimens cured for 28 days was also the highest at an 11% gypsum content (41.8 MPa), but it was close to that at a 14% gypsum content (40.6 MPa). At the age of 60 days, the strength was the highest (46.1 MPa) when the gypsum content was 14%. When the age was extended to 120 days, the strength increased with an increase in gypsum dosage; thus, the ‘critical dosage’ could be regarded as exceeding 20%.

The 7-day compressive strength of the specimens cured at 50 °C was similar to that at 20 °C with an increase in gypsum dosage, first increasing and then decreasing. The strength reached the maximum value of 47.2 MPa when the dosage was 14%. In addition, at 50 °C or 80 °C, the compressive strength increased with an increase in gypsum dosage. The longer the curing age, the more obvious the strength improvement effect. Similarly, the ‘critical dosage’ was not reflected at this time; thus, it can be concluded that the ‘critical dosage’ was above 20%.

### 3.2. Expansion Rate

The horizontal expansion curves of the S1, S3, and S5 specimens along the direction of the copper nail heads under the conditions of 20 °C, 50 °C, and 80 °C are shown in Figure 3. In order to observe the early expansion and final expansion, the expansion curves of 0~480 h were magnified for each age.

As shown in Figure 3, the higher the gypsum content, the greater the expansion rate and the longer the time needed for the expansion to stabilize. The higher the curing temperature, the faster the early expansion and the shorter the time needed for the expansion to stabilize; however, the final expansion rate was lower. In terms of final expansion, the specimens cured at 20 °C had the highest expansion rate, followed by those at 50 °C, and the lowest rates were at 80 °C. At 20 °C curing, the final expansion rate of the specimen increased significantly with the increase in gypsum content, and the time to reach the expansion equilibrium was also greatly delayed. For example, the S1 group with lower (8%) gypsum content reached the equilibrium expansion rate of about 0.1 × 10^−4^ at about 200 h, while the S5 group with 20% gypsum content reached the expansion rate of 2.5 × 10^−4^ at 2880 h, though it still expanded with the extension of age. The variation rule of the expansion rate of the specimens at 50 °C was similar to that at 20 °C, but the expansion rate was decreased. When cured at 80 °C, the expansion rate of the specimens showed little development with curing time, and S5 still had the highest expansion rate.

In comparing the expansion curves of the specimens up to 480 h, it can be seen that S1 still had the highest expansion rate at the maintained temperature of 20 °C (0.09 × 10^−4^), followed by 50 °C (0.06 × 10^−4^); the lowest rate was at the maintained temperature of 80 °C (0.01 × 10^−4^). However, when the gypsum content was increased to 14% (S3), the expansion rate of the specimen at 50 °C was the highest within 200 h. Furthermore, the expansion rate of the specimen at 80 °C was also higher than that at 20 °C within 50 h. With an increase in the gypsum content to 20%, the expansion rates of the specimens at different curing temperatures for 250 h were similar, at about 0.25 × 10^−4^, but they were all much higher than the expansion rates of the specimens with low gypsum content at the same temperatures.

The expansion rate of the specimens indirectly reflected the relatively high content of AFt in the system. The reason for the decrease in the expansion rate of the specimen at 80 °C water curing was due to the difficulty in forming the AFt at a high curing temperature, the thermal decomposition of the AFt, and the dissolution of the AFt at higher temperatures. In general, the RPC reaches the stabilized expansion rate earlier with an increase in the water curing temperature.

### 3.3. X-Ray Diffraction Analysis

Figure 4 presents the results given by the XRD analysis for cement paste with 8%, 14%, and 20% gypsum contents after 7 days and 120 days of curing at different temperatures. The main crystallized phases were ettringite, gypsum, portlandite, and some clinker minerals that were not fully hydrated. When the gypsum content was high, there was an obvious gypsum peak at 7 days. When the amount of gypsum was low, AFm also appeared at 80 °C curing. The CH and clinker peaks weakened, the gypsum peaks gradually disappeared, and the AFt peaks became more intense with an increase in curing temperature, and the same pattern was also detected with prolongation of the curing age.

It can be seen by comparing the S1 and S5 groups of specimens at 120 days of age that the CH peak of the S1 specimens with a lower gypsum dosage was significantly more intense than that of the S5 specimens at all the curing temperatures; laterally, this reflects the fact that the slag in the S5 group had a higher degree of hydration; thus, the corresponding strength was higher. Meanwhile, traces of AFm were detected in the S1 group when cured at 80 °C, while there was no AFm detected in the S5 group. From this, two speculations can be made. Firstly, there may be a thermal decomposition process of AFt at 80 °C. The second speculation is that the rate of hydration was accelerated by high temperature and that the SO_4_^2−^ concentration provided by the gypsum was insufficient. However, under the conditions of sufficient gypsum, AFt can exist in a stable manner.

At high curing temperatures, the C-S(A)-H gel will have a strong adsorption effect on SO_4_^2−^, resulting in a low SO_4_^2−^ concentration in the system, which prevents the generation of AFt. However, in the following curing process, the SO_4_^2−^ is gradually desorbed and continues to react with the AFm to generate AFt. Overall, the existence of AFt during water conditioning at 80 °C is affected by the dynamic equilibrium effect of thermal decomposition and synthesis on the one hand and the SO_4_^2−^ concentration on the other.

### 3.4. TGA Analysis

The TGA analysis results of S1 and S5 at 20, 50, and 80 °C for 7 days and 120 days are shown in Figure 5. The derivative peaks confirm the presence of AFt, AFm, gypsum, CH, and some unreacted slag; these findings are consistent with the XRD results. The main derivative peak in the range of 70–120 °C corresponds to the water removal from ettringite and the decomposition of C-(A)-S-H gel, which is significantly influenced by various curing conditions. The second peak happens in the 100–150 °C temperature range and can be ascribed to gypsum. The third peak is from AFm, which loses its bound water at around 150 °C. The fourth peak in the range of 400–500 °C corresponds to CH. Another peak found in the temperature range of 500–700 °C can be ascribed to the unreacted slag.

At the age of 7 days, the intensity of the AFt peaks was enhanced, and the intensity of the CH peaks was weakened with an increase in the curing temperature and gypsum dosage. Obvious gypsum peaks were observed in S5, and the weight loss decreased with an increase in curing temperature, which was consistent with the XRD results. It can also be seen that there were small AFm peaks from S1 cured at 80 °C. After 120 days of hydration, the trends of the AFt and CH peaks were similar to those at 7 days. The AFm peak was also found in S1 cured at 80 °C. However, the gypsum peak in S5 disappeared.

### 3.5. Scanning Electron Analysis

Figure 6 shows the SEM images of S1, S3, and S5 at different temperatures at the age of 7 days. At 20 °C, a large amount of acicular AFt was observed in the S1 group with low gypsum dosing, and no gypsum was found, whereas both of the S3 and S5 groups with higher gypsum content were observed to have obvious unhydrated gypsum in addition to AFt. At 50 °C, the CH co-existed with AFt in the S1 group, while a large amount of AFt was produced in the S3 and S5 groups; AFt was generated in large quantities, and the morphology was mostly needle-like. At 80 °C, in addition to the presence of AFt, flap-like AFm appeared in the S1 group, while AFt was more obvious and grew into short columns in the S3 and S5 groups.

SEM images of the specimens cured for 120 days are shown in Figure 7. It was observed that the hydration products in the pores of S1 at 20 °C were mainly flaky CH and a small amount of AFt, while the pores of S3 and S5 with elevated gypsum doping were mostly filled with AFt, especially in S3. The hydration products of the samples at 50 °C were similar to those at 20 °C, but the hydration pores in S3 and S5 with the large amount of gypsum doping were further filled. At 80 °C, the hydration products observed in S1 changed considerably, although the pores were also further filled; AFm was present in addition to CH and AFt. The pores in S3 and S5 cured at 80 °C were further filled by AFt, resulting in a denser cement structure.

### 3.6. MIP Analysis

The cumulative pore size distribution (CPSD) and total porosity of S1 and S5 cured at various temperatures, as investigated in this work, are plotted in Figure 8. At 7 d, with an increase in curing temperature, the porosity and the main pore size of S1 and S5 decreased, which indicated that the structure samples were denser. The main pore size of S5 was smaller than that of S1 at all the other temperatures, except at 20 °C. This means that paste with a higher gypsum content has a denser structure at elevated temperatures. However, at 20 °C, the porosity of S5 was larger than that of S1, which may be due to the retarding effect of gypsum. The higher gypsum content of S5 leads to the lower hydration of the paste in an early stage.

When the curing age was up to 120 days, the effect of temperature on the porosity of the specimens was similar to that at 7 days. At the same curing temperature, the total porosity and the main pore size of S5 were less than those of S1. It can be seen that with an increase in gypsum content, the internal structure of the cement is more compact, which corresponds to the SEM results. The porosity of the specimen was reduced by the largest percentage of 38.6% at 80 °C. A possible reason is that at 80 °C, due to the insufficient gypsum in S1, the AFt was poorly stabilized and transformed into AFm, leading to an increase in porosity. The sufficient gypsum in S5 resulted in the AFt being more stable, which not only offset the increase in porosity due to the increase in the density of the C-S-H gel but also filled the pores and reduced the porosity inside the paste.

## 4. Discussion

### 4.1. Hydration and Hardening Process of LHEC

When combined with water, Portland cement clinker hydrates immediately, generating hydration products, such as C-S-H and CH, increasing the alkalinity of the system. Then, the hydration of the slag is greatly activated by the Portlandite (CH) released by the hydration of the clinker, producing Ca^2+^ and AlO^2−^ and reacting with gypsum to produce the AFt and C-S-H [21]. AFt is the main hydration product of LHEC. The AFt crystal clusters are intricately connected to create a framework structure, with the spaces being filled by C-S-H. This results in the cement achieving a high level of density and strength [22], which corresponds to the MIP results and SEM images. Meanwhile, the solid-phase volume increases by approximately 120% during the formation of AFt, giving LHEC a micro-expansion performance [23]. The hydration rate of cement is influenced by the gypsum content and the curing temperature in this process. These factors also affect the formation pace, amount, and stability of AFt, which, in turn, affect the cement’s strength and expansion rate.

### 4.2. Effect of Curing Temperature on Strength and Expansion

The curing temperature has the following impacts on the strength of the specimens: (i) it influences the hydration process of the cement, including the disintegration of the slag glass and the formation of the AFt [24]; (ii) it affects the stability of the AFt [25]; and (iii) it affects the SO_4_^2−^ concentration and the solubility of the AFt [26]. When the curing temperature is below 50 °C, increasing the temperature is favorable for strength. A rise in temperature causes an increase in the rate of slag disintegration and clinker hydration, which, in turn, accelerates the creation of AFt and greatly boosts cement strength. On the other hand, overheating speeds up the formation of AFt but also causes decomposition and increases the binding effect of C-S-H on SO_4_^2−^, making the formation of AFt more difficult and increasing its solubility.

In terms of expansion, raising the curing temperature accelerates the formation of AFt, which raises the early expansion rate. However, the ultimate expansion shrinks, and the stabilization period shortens. The expansion rate depends on the amount of AFt, AFt morphology, collaboration with C-S-H, and strength constraints. The higher the curing temperature, the more AFt is formed in the early phase of early hydration, since the AFt formed in the early stage does not lead to expansion. Relatively less AFt is formed in the later stage, and the strength constraints are stronger, resulting in a lower expansion rate.

### 4.3. Effect of Gypsum Content on Strength and Expansion

Gypsum stimulates the activity of slag, generating AFt, which acts as a structural framework that favors the strength. However, excessive AFt formation may produce expansion, which is detrimental to strength. Meanwhile, gypsum influences the decomposition of AFt or transformation to AFm, which is unfavorable to strength. The effect of the gypsum dosage on strength is a comprehensive effect of the above factors. Low gypsum content means low SO_4_^2−^ concentration, and therefore the low generation of AFt, which is transformed into AFm, leading to a reduction in the strength of the cement. Excessive amounts of gypsum lead to the continued formation and growth of AFt in clusters, resulting in increased expansion stress damage and also unfavorable effects on strength. Thus, there exists a better range of gypsum dosage for the strength of LHEC, which is the reason for the existence of the ‘critical dosage’.

The AFt in LHEC is formed rapidly in the early stage, resulting in a rapid consumption of gypsum. As the hydration proceeds, C-S-H continues to form, repairing the early AFt damage. Therefore, the ‘critical dosage’ becomes larger with an increase in age.

With an increase in curing temperature, the clinker hydration rate and slag disintegration rate increase, accelerating the generation of AFt. This indicates that more AFt is formed during the early stage and that the binding action of C-S-H on SO_4_^2−^ is also strengthened. Therefore, the gypsum ‘critical dosage’ becomes larger. In this study, the compressive strength at 50 °C and 80 °C mainly increased with an increase in gypsum doping; consequently, it can be concluded that the ‘critical dosage’ is greater than 20%. If the curing temperature is too high and the gypsum content is low, the AFt will be transformed into AFm, which is unfavorable to the strength.

At the same curing temperature, the expansion rate of the sample increases with an increase in gypsum content. The higher the amount of gypsum, the more ettringite is produced, resulting in an increased final expansion of the LHEC. An excessive amount of gypsum causes the sample to continue expanding and lengthens the time it takes to stabilize. When the curing temperature is increased, the expansion rate is reduced, and the time for the expansion to stabilize is shortened.

Based on the study above, raising the curing temperature results in a larger ‘critical gypsum dosage’ for LHEC strength.

## 5. Conclusions

In this study, the effect of curing temperature and gypsum content on the strength and expansion of LHEC was investigated. Three temperatures (20 °C, 50 °C, and 80 °C) and five gypsum contents (8%, 11%, 14%, 17%, and 20%) were considered for the cement samples. The main conclusions are summarized as follows:

(1)There is a ‘critical gypsum dosage’ for the flexural and compressive strength at 20 °C, below which the strength increases or changes slightly with an increase in gypsum dosage and above which the strength decreases significantly. With an increase in curing temperature or extension of the curing age, the ‘critical dosage’ becomes larger. A higher curing temperature is more favorable to the strength of specimens with a high gypsum dosage. When the gypsum dosage is too low, a higher curing temperature is not conducive to greater strength.(2)The higher the gypsum content, the greater the expansion rate and the longer the time needed for the expansion to stabilize. The higher the curing temperature, the shorter the time needed for the expansion to stabilize and the lower the final expansion rate.(3)An increase in the gypsum dosage and maintenance temperature promoted the hydration of slag and the formation of AFt and improved the density of the specimens. AFt decomposed at high temperature in the specimens with low gypsum dosing.

In conclusion, LHEC can be applied in a hygrothermal environment by appropriately increasing the gypsum content to ensure the strength and micro-expansion of the cement and a denser internal structure.

## Figures and Tables

**Figure 1 materials-17-05766-f001:**
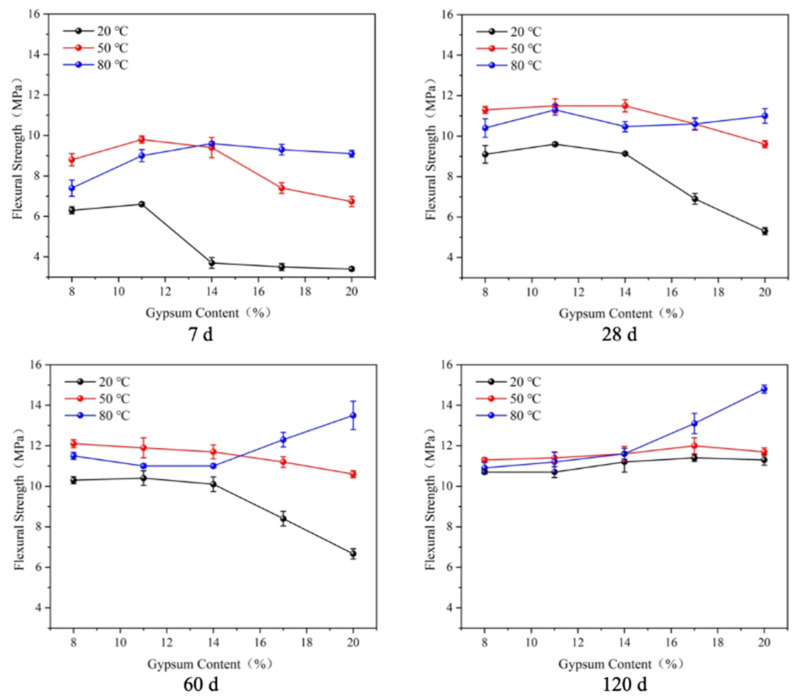
Flexural strength of LHEC.

**Figure 2 materials-17-05766-f002:**
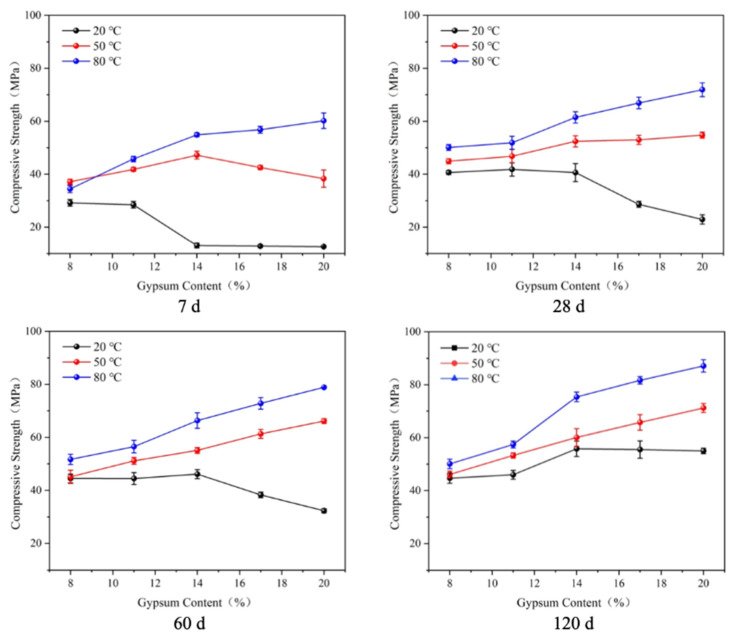
Compressive strength of LHEC.

**Figure 3 materials-17-05766-f003:**
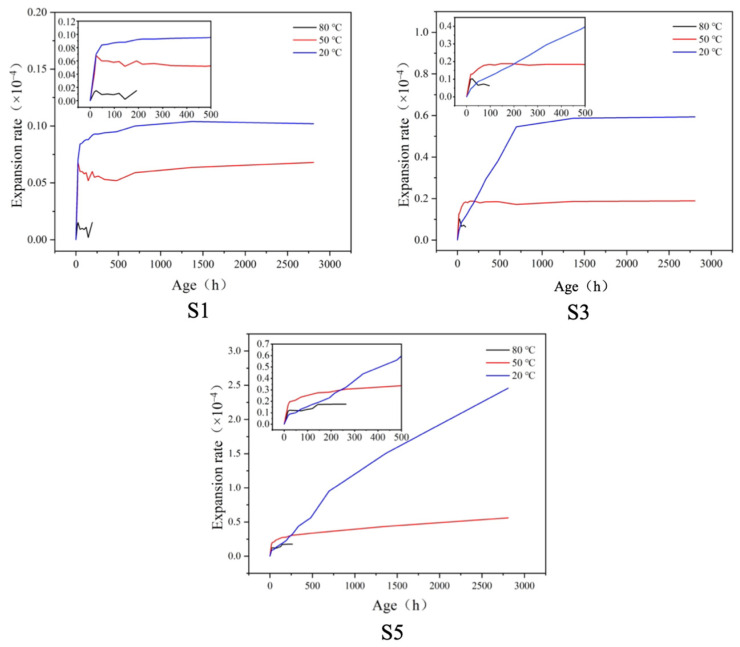
The horizontal expansion rates of S1, S3, and S5.

**Figure 4 materials-17-05766-f004:**
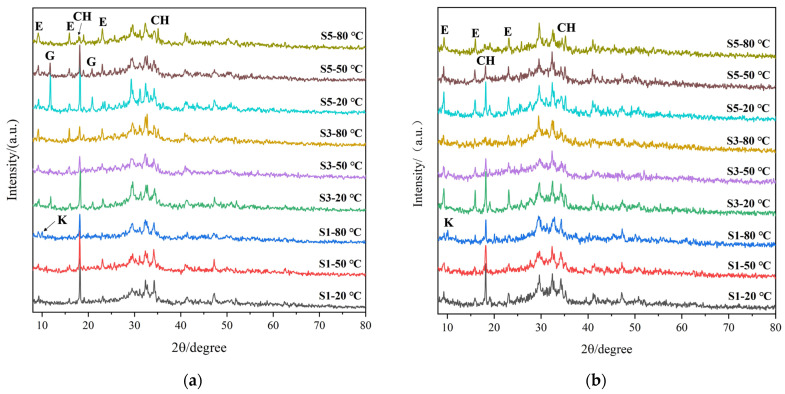
XRD patterns of S1, S3, and S5: (**a**) 7 days; (**b**) 120 days. (E: AFt; CH: Ca(OH)_2_; K: AFm; G: gypsum).

**Figure 5 materials-17-05766-f005:**
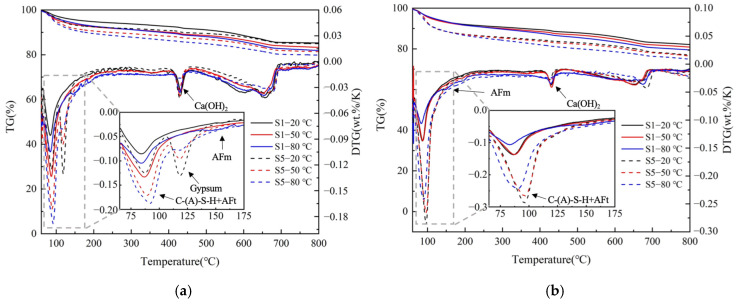
TG profiles of S1 and S5 cured at 20, 50, and 80 °C for (**a**) 7 days; (**b**) 120 days.

**Figure 6 materials-17-05766-f006:**
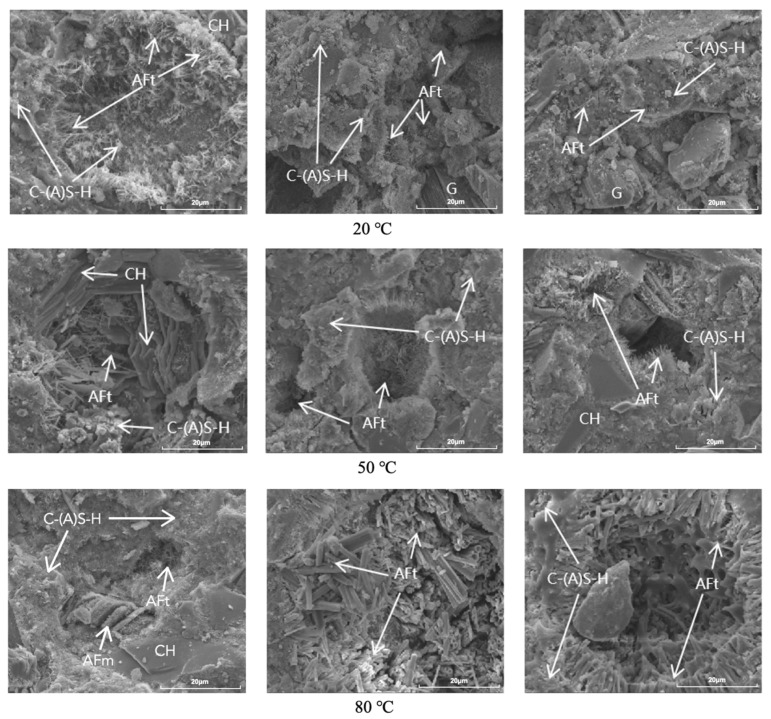
SEM images of S1 (**left**), S3 (**middle**), and S5 (**right**) hydrated for 7 days at different curing temperatures.

**Figure 7 materials-17-05766-f007:**
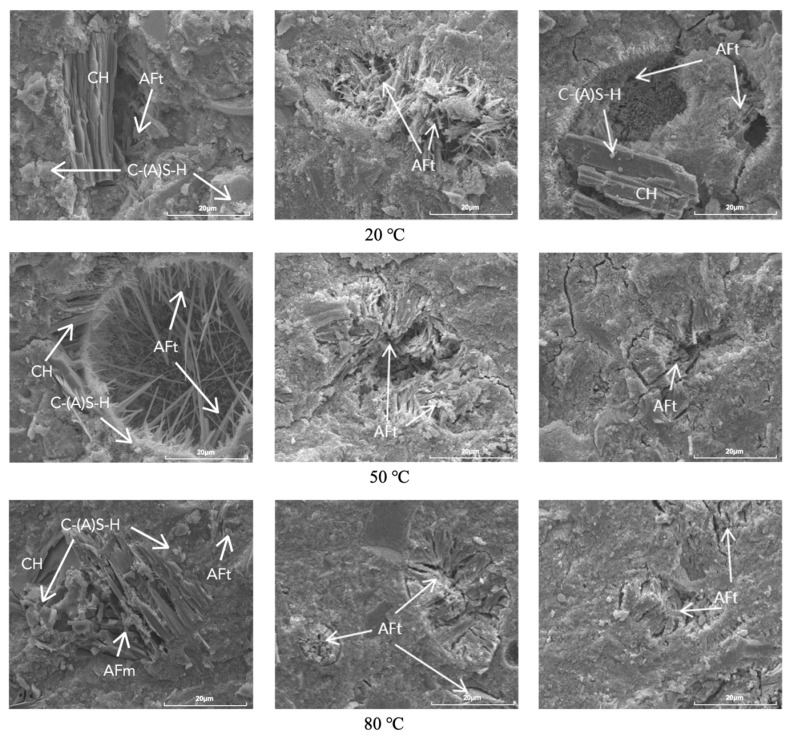
SEM images of S1 (**left**), S3 (**middle**), and S5 (**right**) hydrated for 120 days at different curing temperatures.

**Figure 8 materials-17-05766-f008:**
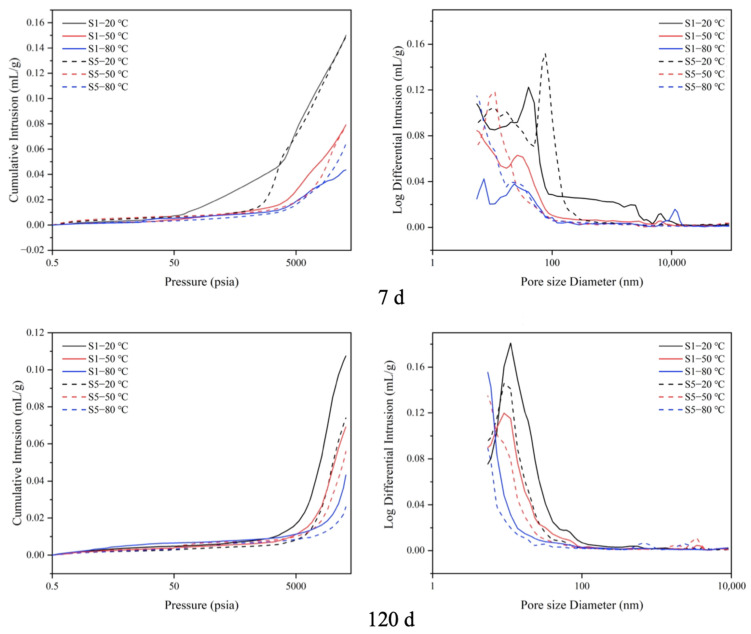
Porosity (**left**) and cumulative pore size distribution (**right**) of S1 and S5 cement pastes measured by MIP at 7 and 120 days.

**Table 1 materials-17-05766-t001:** Chemical composition of raw materials (wt%).

	*LOI	SiO_2_	Al_2_O_3_	Fe_2_O_3_	CaO	MgO	Na_2_O	SO_3_
Clinker	/	22.04	4.51	3.30	64.66	2.90	0.60	0.46
GBFS	/	34.44	15.60	1.35	39.48	6.76	/	0.21
Gypsum	22.71	2.18	0.91	0.49	30.07	0.25	/	42.38

* LOI: loss of ignition.

**Table 2 materials-17-05766-t002:** Chemical composition of raw materials (wt%).

Sample Label	Clinker	GBFS	Gypsum
S1	32	60	8
S2	32	57	11
S3	32	54	14
S4	32	51	17
S5	32	48	20

## Data Availability

The data presented in this study are not available due to privacy.

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
