# Peer review of "Strength and Expansion of LHEC with Different Gypsum Contents Under Thermal Curing"

_materials, 2024, doi:10.3390/ma17235766_

Round 1

Reviewer 1 Report

Comments and Suggestions for Authors

General concept comments

The weakness is the lack of a formulated scientific hypothesis. Accordingly, it isn't easy to assess the hypothesis's testability.

The data should be compared in detail with known analogues throughout the manuscript.

Specific comments

 In subsection 2.1, it is necessary to indicate which phase of gypsum was used (number of water molecules).

In Table 1, it should be probably specified as Loss of Ignition (LOI) instead of Loss.

In subsection 2.2, the relative humidity in the curing box at t=20 °C should be specified.

There is an error in the value of the denominator in formula (1). The value should be equal to L0, i.e. 280 mm.

Figure 3 and its description in the text should indicate the dimension of the expansion rate.

General questions

Given the need to analyse the current state of the problem, it can be recommended to change the references that are older than 5 years. It is also strongly recommended to change the references #2, #3, #4,#5, #11, #14, #15, #16, #17, #20, #21, #24, #25, and #26 in any case because of obsolescence. The reference #3 is in incorrect format. The relevance of reference #16 is difficult to assess given the language.

It is inappropriate to refer to several papers in one sentence. Thus, the references [5-7] and [12-15] should be separated by a comment, at least one sentence per each cited publication.

Author Response

Comment 1: The weakness is the lack of a formulated scientific hypothesis. Accordingly, it isn't easy to assess the hypothesis's testability. The data should be compared in detail with known analogues throughout the manuscript.

Response 1: Thank you for your comments. Before the experiment, we assume that the incorporation of gypsum may enhance the strength of LHEC curing at high temperature by increase the stability of AFt. As suggested by the reviewer, a hypothesis has been added in the revised manuscript, as seen in page 3 line 118~120. In the original manuscript, we mentioned about quality problems such as strength shrinkage of ordinary Portland cement-based materials in high temperature environment, as seen in page 2, line 68~69. This is in contrast to the strength performance of LHEC curing at high temperature in this paper. Gypsum is a critical component of LHEC, which can be seen as an exciter of slag activity, promoting the disintegration and hydration of slag. Thus, we focused on the focuses on the comparison of LHEC in high temperature environment and ambient environments to investigate the influence of gypsum content on LHEC. In the future, the authors would pay more attention to improve the experimental prototype.

Comment 2: In subsection 2.1, it is necessary to indicate which phase of gypsum was used (number of water molecules).

Response 2: Thank you for pointing this out. We agree with this comment. In this experiment, we used desulfurization gypsum with two crystal waters (CaSO4·2H2O). Thus, the phase of gypsum we used has been indicated in the revised manuscript, as seen in page 3 line 130.

Comment 3: In Table 1, it should be probably specified as Loss of Ignition (LOI) instead of Loss.

Response 3: We are really sorry for our careless mistakes. Thank you for your reminder. As suggested by the reviewer, we have corrected the “Loss” into “LOI”, as seen in page 3 line 137.

Comment 4: In subsection 2.2, the relative humidity in the curing box at t=20 °C should be specified.

Response 4: The authors apologize for confusing the reviewer that the hardened specimens were cured in water in cement constant temperature and humidity standard curing box at 20 ℃. In the original manuscript, the context about “in water” is at the end of the sentence. To give a clearer understanding, in the revised manuscript, the correspondent context has been reorganized, which can be found in page 4 line 145.

Comment 5: There is an error in the value of the denominator in formula (1). The value should be equal to L0, i.e. 280 mm.

Response 5: Thank you for your careful checks. We are sorry for our carelessness. Based on your comments, we have made the corrections in page 4 line 162.

Comment 6: Figure 3 and its description in the text should indicate the dimension of the expansion rate.

Response 6: Thank you for pointing this out. The authors totally agree with the reviewer. We tested the horizontal expansion rate along the copper heads direction. In the revised manuscript, the supplementary description has been updated in the text and figure 3, which can be found in page 7 line 239~240 and page 8 line 258.

Comment 7: Given the need to analyse the current state of the problem, it can be recommended to change the references that are older than 5 years. It is also strongly recommended to change the references #2, #3, #4, #5, #11, #14, #15, #16, #17, #20, #21, #24, #25, and #26 in any case because of obsolescence. The reference #3 is in incorrect format. The relevance of reference #16 is difficult to assess given the language.

Response 7: We sincerely appreciate the valuable comments. We are really sorry for our careless mistakes. We have checked the literature carefully and changed the references that are older than 5 years, which can be found in page 15 line 478~533.

Comment 8: It is inappropriate to refer to several papers in one sentence. Thus, the references [5-7] and [12-15] should be separated by a comment, at least one sentence per each cited publication.

Response 8: Thank you for your comments. We agree with this comment. The original references [5], [6], [7] and [12], [13], [14], [15] supported one sentence, so we have deleted the extra references, kept one cited publication for each sentence. In the revised manuscript, the correspondent context has been modified in page 2 line 61, 77, 79.

Reviewer 2 Report

Comments and Suggestions for Authors

The manuscript presents an experimental study on the performance of Low Heat-Slight Expansive Cement (LHEC) under hygrothermal conditions. The study provides data on the strength and expansion behavior of LHEC with varying gypsum dosages (8% - 20%) at 20 ℃, 50 ℃, and 80 ℃. The incorporation of gypsum into Portland cement is known to reduce strength and increase porosity due to temperature increases. However, in LHEC systems, gypsum promotes slag disintegration and hydration, enhancing LHEC performance in hot and humid environments. Another crucial aspect of the study is its contribution to reducing carbon emissions in the cement industry and promoting sustainable societal development. The results are thoroughly discussed, particularly with respect to the X-ray Diffraction Analysis (XRD), which provides scientific justification for the empirical findings.

The paper is interesting and well-organized. Nevertheless, certain sections could be improved for clarity and precision. The authors are encouraged to consider the following suggestions before submitting a revised version:

Introduction:

The first part of the introduction is well-discussed, but it becomes tedious in later sections. The part discussing the sustainability challenges of concrete production lacks references. The reviewer recommends considering the following works: https://doi.org/10.1016/j.conbuildmat.2020.122124 and https://doi.org/10.3390/ma17143408.

Overall Text:

The language requires refinement to improve readability and correct errors, such as the typo in line 112 where "appled" should be "applied," and the labeling error in Figure 1 where "Ggypsum" should be corrected. Additionally, some parts of the text lack clarity and coherence.

Rephrase the sentence in lines 196-1981: “Noticeably, although the strengths of specimens under 50 ℃ and 80 ℃ were higher than that of specimens under 20 ℃, it is not the case that the higher the temperature was, the better the flexural strength of specimens was.”

Section 2. Experimental Program:

·        Clarify how many samples were manufactured for each test.

·        Include images of the experimental tests, sample preparation, and observed failure modes to enhance the section's descriptive quality.

Section 2.2. Strength Test:

·        Cite the relevant standards, specifically "GB/T 17671-1999" on line 134.

·        Explain abbreviations clearly, such as "for 7 d, 28 d, 60 d, and 120 d" in line 139, to improve comprehensibility.

Section 3. Results:

·        It is recommended to plot Figures 1 and 2 using the same range on the y-axes for easier comparison.

·        Consider normalizing the results relative to the reference mix (without gypsum) for better evaluation.

·        Specify if the results in Figures 1 and 2 are averages, and provide the coefficient of variation for each data set.

·        Figure 6 is not referenced in the text—please ensure that it is cited appropriately.

Section 4. Discussion:

·        The statement, “This results in the cement achieving a high level of density and strength,” is unsupported. To make this claim regarding LHEC with gypsum, results should be compared with a reference mix without gypsum (0%).

·        Why are the mechanical properties of the reference mix not reported in the paper? Including them would strengthen the conclusions.

References:

·        Review and adjust the references to ensure they adhere to the journal's formatting guidelines.

·        Specifically, check and correct references [2] and [3] for accuracy and consistency.

Comments on the Quality of English Language

Although I am not qualified to assess the quality of English in this paper, some parts seem full of errors and lack comprehensiveness.

Author Response

Comment 1: The first part of the introduction is well-discussed, but it becomes tedious in later sections. The part discussing the sustainability challenges of concrete production lacks references. The reviewer recommends considering the following works: https://doi.org/10.1016/j.conbuildmat.2020.122124 and https://doi.org/10.3390/ma17143408.

Response 1: We sincerely appreciate the valuable comments. The authors totally agree with the reviewer that references “Influence of Silica Fume, Metakaolin & SBR Latex on Strength and Durability Performance of Pervious Concrete” and “Integrating Plastic Waste into Concrete: Sustainable Solutions for the Environment” discuss methods to increase sustainability in the concrete industry. In the revised manuscript, they have been added as references [2] and [3], as seen in page 1 line 43~47. Meanwhile, the other related references have also been referred.

Comment 2: The language requires refinement to improve readability and correct errors, such as the typo in line 112 where "appled" should be "applied," and the labeling error in Figure 1 where "Ggypsum" should be corrected. Additionally, some parts of the text lack clarity and coherence.

Response 2: We are really sorry for our careless mistakes. Thank you for your reminder. As suggested by the reviewer, we have corrected the “appled” into “applied” and “Ggypsum” into “Gpsum”, as seen in page 3 line 118 and page 6 line 203. And figure 1 in original manuscript have been checked carefully.

Comment 3: Rephrase the sentence in lines 196-1981: “Noticeably, although the strengths of specimens under 50 ℃ and 80 ℃ were higher than that of specimens under 20 ℃, it is not the case that the higher the temperature was, the better the flexural strength of specimens was.”

Response 3: We thank the reviewer for suggesting to polish the language. We tried our best to improve the manuscript. We polished the language by editing service to improve readability of the manuscript. The sentence you mentioned was reorganized in page 6 line 207~210

Comment 4: Clarify how many samples were manufactured for each test.

Response 4: Thank you for your comments. We think this is an excellent suggestion. In this experiment, we manufactured 3 specimens for each flexural strength and reported average flexural strength. The compressive strength is determined on the broken halves of the specimens. The reported value of compressive strength is the mean of six tests. We have clarified the number of samples we manufactured for each test in page 4 line 149~151.

Comment 5: Include images of the experimental tests, sample preparation, and observed failure modes to enhance the section's descriptive quality.

Response 5: Thank you for pointing this out. Since there is standard of the method of experimental tests, we don’t show the images of experimental tests in text due to the limit of space. We have added images of experimental tests including specimens before demolding, specimens after demolding and broken specimens after flexural strength test in supplementary file.

Comment 6: Cite the relevant standards, specifically "GB/T 17671-1999" on line 134.

Response 6: The authors agree with this comment. Therefore, we have cited the relevant standard “GB/T 17671-2021” in page 4 line 142.

Comment 7: Explain abbreviations clearly, such as "for 7 d, 28 d, 60 d, and 120 d" in line 139, to improve comprehensibility.

Response 7: Thank you for your careful checks. For simplicity, we use “7 d, 28 d, 60 d and 120 d” present the specimens curing for 7, 28, 60 and 120 days. This is a relatively common abbreviation in many articles, such as following works:

 https://doi.org/10.1016/j.cemconres.2018.03.012    and https://doi.org/10.1016/j.jobe.2023.107519

Comment 8: It is recommended to plot Figures 1 and 2 using the same range on the y-axes for easier comparison.

Response 8: Thank you for your comments. We have modified figure 1 and 2 using the same range on y-axes, which can be found in page 6 line 204 and page 7 line 236.

Comment 9: Consider normalizing the results relative to the reference mix (without gypsum) for better evaluation.

Response 9: Thank you for your comments. We appreciate your suggestion regarding adding up specimens without gypsum as reference mix. However, gypsum is a necessary component used as a retarder in cement. In the LHEC system, gypsum is a key component, which not only used as a retarder, but also can be seen as an exciter of slag activity, promoting the disintegration and hydration of slag. The strength of specimens without gypsum will be poor. Thus, this experiment investigates the influence of gypsum on the strength of specimens in a certain range. The control group without gypsum has little reference value for this experiment.

Comment 10: Specify if the results in Figures 1 and 2 are averages, and provide the coefficient of variation for each data set.

Response 10: We sincerely appreciate the valuable comments. The results in figures 1 and 2 in original manuscript are averages of 3 specimens, we have clarified in page 4 line 149~151. We have added error bars to these 2 figures, which can be found in page 6 line 204 and page 7 line 236.

Comment 11: Figure 6 is not referenced in the text—please ensure that it is cited appropriately.

Response 11: We are really sorry for our careless mistakes. Thank you for your reminder. We have cited figure 6 in original manuscript which is figure 7 in revised manuscript in page 10 line 316.

Comment 12: The statement, “This results in the cement achieving a high level of density and strength,” is unsupported. To make this claim regarding LHEC with gypsum, results should be compared with a reference mix without gypsum (0%).

Response 12: We sincerely appreciate the valuable comments. We added a conference to support the statement of the hydration of slag can increase the density and strength of cement, as seen in page 13 line 375. LHEC has a requirement for gypsum content. Thus, we compared the performance of specimens with different gypsum contents. According to the results of strength and porosity, the increase of gypsum content can improve the strength and porosity.

Comment 13: Why are the mechanical properties of the reference mix not reported in the paper? Including them would strengthen the conclusions.

Response 13: Thank you for your comment. In the LHEC system, gypsum is a key component, which not only used as a retarder, but also can be seen as an exciter of slag activity, promoting the disintegration and hydration of slag. The strength of specimens without gypsum will be poor. Thus, we take specimens with low gypsum content rather than specimens without gypsum as reference mix.

Comment 14: Review and adjust the references to ensure they adhere to the journal's formatting guidelines.Specifically, check and correct references [2] and [3] for accuracy and consistency.

Response 14: Thank you for your careful checks. We are really sorry for our careless mistakes. We have checked the literature carefully and adjust the references to ensure they adhere to the journal's formatting guidelines., which can be found in page 15.

Round 2

Reviewer 2 Report

Comments and Suggestions for Authors

The revised version of the paper is complete, and this reviewer accepts it for publication in the current form.